# Age-related changes in the hematopoietic stem cell pool revealed via quantifying the balance of symmetric and asymmetric divisions

Teiko Kawahigashi[1☯], Shoya Iwanami[2☯], Munetomo Takahashi[3,4], Joydeep Bhadury[5], Shingo Iwami[2,6,7,8,9,10]*, Satoshi Yamazaki[1,11]*

1 Division of Stem Cell Biology, Center for Stem Cell Biology and Regenerative Medicine, The Institute of Medical Science, University of Tokyo, Tokyo, Japan, 2 interdisciplinary Biology Laboratory (iBLab), Division of Natural Science, Graduate School of Science, Nagoya University, Nagoya, Japan, 3 Graduate School and Faculty of Medicine, The University of Tokyo, Tokyo, Japan, 4 Medical Research Council Toxicology Unit, Gleeson Building, Tennis Court Road, University of Cambridge, Cambridge, United Kingdom, 5 Institute for Stem Cell Biology and Regenerative Medicine, Stanford University School of Medicine, Stanford, CA, United States of America, 6 Institute of Mathematics for Industry, Kyushu University, Fukuoka, Japan, 7 Institute for the Advanced Study of Human Biology (ASHBi), Kyoto University, Kyoto, Japan, 8 Interdisciplinary Theoretical and Mathematical Sciences Program (iTHEMS), RIKEN, Saitama, Japan, 9 NEXT-Ganken Program, Japanese Foundation for Cancer Research (JFCR), Tokyo, Japan, 10 Science Groove Inc., Fukuoka, Japan, 11 Laboratory of Stem Cell Therapy, Faculty of Medicine, University of Tsukuba, Tsukuba, Japan

☯ These authors contributed equally to this work.
* iwami.iblab@bio.nagoya-u.ac.jp (SI); y-sato4@md.tsukuba.ac.jp (SY)

**Data Availability Statement:** All relevant data are within the paper and its Supporting Information files.

## Abstract

Hematopoietic stem cells (HSCs) are somatic stem cells that continuously generate lifelong supply of blood cells through a balance of symmetric and asymmetric divisions. It is well established that the HSC pool increases with age. However, not much is known about the underlying cause for these observed changes. Here, using a novel method combining single-cell *ex vivo* HSC expansion with mathematical modeling, we quantify HSC division types (stem cell—stem cell (S-S) division, stem cell—progenitor cell (S-P) division, and progenitor cell—progenitor cell (P-P) division) as a function of the aging process. Our time-series experiments reveal how changes in these three modes of division can explain the increase in HSC numbers with age. Contrary to the popular notion that HSCs divide predominantly through S-P divisions, we show that S-S divisions are predominant throughout the lifespan of the animal, thereby expanding the HSC pool. We, therefore, provide a novel mathematical model-based experimental validation for reflecting HSC dynamics *in vivo*.

## Introduction

Hematopoietic stem cells (HSCs) are special blood cells that can self-renew and differentiate into every cell type within the blood system throughout the organism's lifespan [1–5]. In mouse models, HSCs change quantitatively and functionally as the animals ages [6, 7]. These

**Funding:** The author(s) received no specific funding for this work.

**Competing interests:** The authors have declared that no competing interests exist.

age-associated changes in HSCs have been functionally validated using transplantation assays [8–12], but quantitative assessment of the underlying kinetics has been limited. To simultaneously maintain the HSC pool and differentiated progenies, HSCs must regulate their cell division in response to environmental cues [13–16]. Asymmetric cell divisions (one self-renewing daughter cell and one differentiated daughter cell) are known to maintain tissue homeostasis [17]. Recently, there is growing evidence that tissue homeostasis can also be maintained by symmetric divisions within the stem cell pool [13–16]. Taken together, these results suggest that HSCs undergo three types of divisions to maintain tissue homeostasis: symmetric self-renewal (stem cell-stem cell division, S-S division), asymmetric self-renewal (stem cell-progenitor cell division, S-P division), and symmetric differentiation (progenitor cell-progenitor cell division, P-P division)7. Although absolute HSC numbers are known to increase with age [6, 7, 18, 19], the cell division mechanisms governing these changes remain largely unknown.

The *ex vivo* expansion of HSCs has been one of the long-standing issues in hematology. Long-term maintenance and expansion of HSCs *ex vivo* provide opportunities to better assess stem cell function and improve therapeutic use. However, finding the correct culture conditions to achieve this has been challenging for decades. We recently found that murine HSCs could be efficiently expanded in polyvinyl alcohol (PVA)-containing culture medium, enabling between 236- and 899-fold expansions of functional HSCs over one month20. Our novel culture system allows us to reproduce *in vivo* environments *ex vivo* better than before, offering a novel approach to address previously unresolved issues in hematology.

Here, by using our novel single-cell HSCs expansion system [20] in combination with mathematical modeling, we show how age-related changes in the HSCs division can be explained by a balance of symmetric and asymmetric divisions. Our quantitative mathematical modeling enables the inference of hematopoietic stem and progenitor cell population sizes of mice at different ages, laying the framework to understand the mechanisms behind clonal hematopoiesis.

## Materials and methods

### Mice

C57BL/6-CD45.2 and C57BL/6-CD45.1 (PepboyJ) mice were purchased from Japan SLC, Sankyo-Lab Service, or bred in-house. 6–72-week-old male mice were used for the analysis. All mice were housed in a specific pathogen-free (SPF) condition with free access to food and water. Mice were sacrificed by cervical dislocation after isoflurane anesthesia. All animal protocols were approved by the Animal Care and Use Committee of the Institute of Medical Science University of Tokyo. All efforts were made to minimize suffering.

### Cell collection by fluorescent-activated cell sorting (FACS)

Mouse BM cells were isolated from the femur, pelvis, and sternum, stained with APC-cKit antibody and cKit-positive cells enriched using anti-APC magnetic beads and LS columns (Miltenyi Biotec). cKit-enriched cells were then stained with a lineage antibody cocktail (biotinylated-CD4, -CD8, -CD45RA/B220, -TER119, -Gr1, and -CD127), before being stained with anti-CD34, anti-cKit, anti-Sca1, and streptavidin-APC/eFluor 780 for 90 minutes. Where indicated, cells were also stained with anti-CD150. Cell populations were then purified using a FACS Verse (BD, USA) by direct sorting into wells containing media using PI as a live/dead stain. The antibodies used are shown in S3A Table.

## Serum albumin-free mouse cell cultures

HSCs were cultured in media composed of F12 media, 1% ITSX, 1%, 10 mM HEPES, 1% P/S/G, 100 ng/ml mouse TPO, 10 ng/ml mouse SCF and 0.1% of polyvinyl alcohol (PVA; P8136, 363081, or 363146), at 37°C with 5% CO2. For long-term cultures in PVA-based cultures, media was completely changed every 2–3 days after the first 5–6 days by manually removing conditioned media by pipetting and replacing fresh media as indicated. In 96-well plate wells containing 200 μl of media, this involved gentle removal of 190–200 μl of the conditioned media using a pipette to avoid disturbing cells that were lightly adherent to the well bottom, and then gently pipetting in 200 μl of pre-warmed and freshly-prepared media down the side of the well to minimize disturbing the cell layer. Any cells removed from the well in the conditioned media were discarded. Cultures were performed using 96-well round-bottomed plates. Cell culture was carried out for 14 days.

## Analysis of cultured cells

Following a 14-day ex vivo culture, cells were counted (using a hemocytometer, a CYTORE-CON cytometer, or a Nucleocounter NC-3000). For flow cytometric analysis, cells were stained with a lineage cocktail (CD4, CD8, CD45RA/B220, TER119, Gr-1, and CD127), anti-CD34, anti-cKit, anti-Sca1, anti-CD150, anti-CD201, anti-CD11b antibodies (the antibodies used are shown in S3B Table) for 30–90 minutes. Following a wash step, flow cytometric analysis was performed using a FACS Verse (BD, USA) using PI as a live/dead stain.

## Analysis of age-related murine HSC number

For analysis of age-related changes HSC number, whole bone marrow and skeletal bones (femurs, pelvic bones, and sternums) freshly harvested from mice between 6 to 25 weeks of age, crushed and then stained with lineage antibody cocktail (CD4, CD8, CD45RA/B220, TER119, Gr1, 11b, and CD127), anti-CD34, anti-cKit, anti-Sca1, anti-CD150 antibodies for 30–90 minutes (the antibodies used are shown in S3C Table). three male mice were used for every time point. Following a wash step, flow cytometric analysis was performed using a FACS Verse (BD, USA) using PI as a live/dead stain.

## Cell cycle assay using BrdU

To investigate the turnover rate of CD34−KSL cells, BrdU (Sigma-Aldrich) was administered continuously to mice via drinking water (0.5 mg/ml). After 1 week, CD34−KSL cells were isolated and stained with FITC-conjugated anti-BrdU antibody (Molecular Probes). The cells then were analyzed by ArrayScan.

## Statistical analysis

Results are reported as the mean ± s.d. or the median, unless stated otherwise. All statistical analyses were performed using GraphPad Prism9 (GraphPad). Data were analyzed using one-way analysis of variance followed by a post hoc test, as indicated in the figure legends. The threshold for statistical significance was set to $p < 0.05$.

## Estimation of distributions of proportion of CD201⁺CD150⁺ KSL population

To quantitatively describe the self-renewal ability of cells obtained from each-old mouse, we employed the probability density function of the beta distribution to analyze the distribution of the proportion of CD201⁺CD150⁺KSL population, $x$, at age $t$ with shape parameters, $\alpha_t$ and

$\beta_t$, as follows:

$$f(x, t; \alpha_t, \beta_t) = \frac{x^{\alpha_t - 1}(1 - x)^{\beta_t - 1}}{B(\alpha_t, \beta_t)},$$ (1)

where $B(\alpha_t, \beta_t)$ denotes the beta function. The shape parameters of **Eq (1)** for each age, 4, 5, 6, 7, 8, 9 15, 20, 22, 48, and 52 weeks, were obtained by maximum likelihood estimation. If data values of the proportion of CD201$^+$CD150$^+$KSL population were 0 or 1, they were assumed to be 0.001 or 0.999, respectively, to avoid the inability to calculate likelihood. The estimated values of shape parameters for each age are summarized in **S1 Table**.

## Calculating age-dependent distributions of proportion of CD201$^+$CD150$^+$ KSL population

To predict the age-dependent distribution of CD201$^+$CD150$^+$KSL cell proportions, we assumed the following beta distribution instead of **Eq (1)**:

$$f(x, t; \alpha(t), \beta(t)) = \frac{x^{\alpha(t) - 1}(1 - x)^{\beta(t) - 1}}{B(\alpha(t), \beta(t))}.$$ (2)

Here $\alpha(t)$ and $\beta(t)$ are the shape parameters of the beta distribution depending on age $t$, and given as follows:

$$\alpha(t) = b_{1,\alpha}\exp(-a_{1,\alpha}(t - t_0)) + b_{2,\alpha}\exp(-a_{2,\alpha}(t - t_0)),$$ (3)

$$\beta(t) = b_{1,\beta}\exp(-a_{1,\beta}(t - t_0)) + b_{2,\beta}\exp(-a_{2,\beta}(t - t_0)),$$ (4)

where $t_0$ denotes the age when the distribution of the CD201$^+$CD150$^+$ cell population was first measured in our experiments, that is, $t_0 = 4$. The parameters $a_{1,\alpha}$, $a_{2,\alpha}$, $b_{1,\alpha}$, $b_{2,\alpha}$, $a_{1,\beta}$, $a_{2,\beta}$, $b_{1,\beta}$ and $b_{2,\beta}$ determine the biphasic decay profile of $\alpha(t)$ and $\beta(t)$. From the estimated shape parameters, $\alpha_t$ and $\beta_t$, for 4, 5, 6, 7, 8, 9 15, 20, 22, 48, and 52 weeks of age (see **S1 Table**), the above parameters in **Eqs (3 and 4)** were estimated by a nonlinear least squares method (**S1 Fig**). The estimated parameter values of **Eqs (3 and 4)** are summarized in **S2 Table**.

## Mathematical model of HSC differentiation with the age-dependent HSC divisions

We developed the mathematical model describing the cell differentiation dynamics of HSCs, defined as CD34$^-$CD150$^+$KSL cell population, and progenitor cells as follows:

$$\frac{dS(t)}{dt} = rp(t)S(t) - r(1 - p(t) - q(t))S(t),$$ (5)

$$\frac{dP(t)}{dt} = rq(t)S(t) + 2r(1 - p(t) - q(t))S(t) - dP(t).$$ (6)

where $S(t)$ and $P(t)$ correspond to the number of HSCs and progenitor cells at age $t$, respectively. The sum of the number of HSC and progenitor cells, $S(t)+P(t)$, is assumed to be the number of KSL cells. The parameter $r$ and $d$ represent the rate of cell division of HSCs and differentiation rate of progenitor cells, respectively (**Fig 4A**). We assumed the HSC proliferation rate, $r$, to be constant regardless of age (**Fig 3D**). The differentiation rate, $d$, was fixed to be 0.1 per week (i.e, the mean life is 10 weeks = 70 days) because the differentiation rate of hematopoietic progenitors was reported as 0.008 per day (i.e., the mean life is 125 days)

for leukaemic progenitors in human, and 0.01 and 0.02 per day (i.e., the mean lives are 100 and 50 days) for short-term HSC and multipotent progenitor cells in mice from modeling studies [21, 22]. The sensitivity analysis of the parameter $d$ were given in **S3B–S3E Fig**. The death of HSCs is ignored because the frequency of dead cells in the HSC population was negligible (**Fig 3E**).

The fractions of each cell division type, S-S, S-P, and P-P divisions, at age $t$ are defined as $p(t)$, $q(t)$ and $1-p(t)-q(t)$, respectively (**Fig 4A**). Under the assumption that the distribution of $CD201^+CD150^+KSL$ cell proportion is an indicator of the fraction of cell division types, we derived

$$p(t) = 1 - P_{c_2}(\alpha(t), \beta(t)), \tag{7}$$

$$q(t) = P_{c_2}(\alpha(t), \beta(t)) - P_{c_1}(\alpha(t), \beta(t)). \tag{8}$$

Here we defined

$$P_{c_i}(\alpha(t), \beta(t)) = \sum_{x_i=0}^{c_i-0.1} \hat{f}(x_i < x < x_i + 0.1, t; \alpha(t), \beta(t)), \tag{9}$$

which describes the cumulative distribution function of the following discretized beta function of **Eq (2)** with range $0 \le x \le c_i$ ($c_i = c_1$ or $c_2$ are increments of 0.1), and

$$\hat{f}(x_i < x < x_i + 0.1, t; \alpha(t), \beta(t)) = I_{x_i+0.1}(\alpha(t), \beta(t)) - I_{x_i}(\alpha(t), \beta(t)), \quad x_i$$
$$= 0, 0.1, 0.2, \ldots, 0.9. \tag{10}$$

where $I_*(\alpha(t), \beta(t))$ is the cumulative distribution function of **Eq (2)**. We note that, if we use the continuous beta function, **Eq (2)**, unrealistic small estimated values of $c_1$ may be possible because the very small $c_1$ returns a fraction of P-P differentiation as some magnitude. In fact, our group showed that, when the fraction of $CD201^+CD150^+KSL$ population was even around 10%, the post-transplantation chimerism in peripheral blood became almost $0$[21], implying P-P divisions are still possible at least for 10% of the fraction and supporting our assumption on usage of the discretized beta function.

## Data fitting of the mathematical model to the time-course data of HSCs and KSL cells

The optimal values of $r$, $S(t_0)$ and $P(t_0)$ where $t_0 = 4$ were estimated from data fitting to the time-course data of HSCs and KSL cells by a non-linear least squares method (constrOptim function in R) and the values of sum of squared-residuals (SSR) compared for all combinations of $c_1$ and $c_2$. With the optimal values of $c_1$ and $c_2$, which returned smallest SSR with optimal $r$, $S(t_0)$ and $P(t_0)$, the posterior distributions of $r$, $S(t_0)$ and $P(t_0)$ were estimated by Bayesian inference implemented in Markov chain Monte Carlo methods (modMCMC function in R). The estimated parameter values are summarized in **Table 1**.

For sensitivity analysis, we also investigated $d = 0.05$ per week (i.e., the mean life is 20 weeks = 140 days: 2-folds longer) and 0.2 (i.e., the mean life 5 weeks = 35 days: 2-folds shorter). We confirmed the goodness of fits was comparable with $d = 0.1$ (the values of SSR, were 1.76, 1.19 and 2.91 for $d = 0.1$, 0.05, 0.2, respectively, with the mean values of parameters) (See **S3B–S3E Fig**). We also confirmed similar or slightly higher SSR for $d$ ranging 0.01 to 0.2.

**Table 1. Estimated parameters defining the cell differentiation dynamics of HSCs and progenitor cells.**

| Parameter name | Symbol | Unit | Value |
|---|---|---|---|
| Threshold of the proportion of CD201⁺CD150⁺KSL for P-P and S-P division | $c_1$ | – | 0.3 |
| Threshold of the proportion of CD201⁺CD150⁺KSL for S-P and S-S division | $c_2$ | – | 0.8 |
| Cell proliferation rate | $r$ | week⁻¹ | 0.172 (95% CI: 0.0402–0.330) |
| Differentiation rate | $d$ | week⁻¹ | 0.1* |
| Initial value of HSCs | $S(0)$ | cells/mL | $5.14\times10^3$ (95% CI: $1.13\times10^3$–$1.61\times10^4$) |
| Initial value of progenitor cells | $P(0)$ | cells/mL | $1.01\times10^5$ (95% CI: $5.98\times10^3$–$3.76\times10^5$) |

*Fixed with sensitivity analysis with different values of 0.05 and 0.2.

P-P: Progenitor cell-progenitor cell. S-P: Stem cell-progenitor cell. S-S: Stem cell-stem cell. CI: Credible interval.

## Results and discussion

### Age-related bimodal changes in functional HSC proportions

We performed single-cell expansion by sorting CD34⁻/lowCD150⁺c-Kit⁺Sca-1⁺ lineage marker negative (CD34⁻CD150⁺KSL) cell populations that are highly enriched in functional HSCs [23–25] from mice (Fig 1A). With our HSC expansion system, although we achieve > 200-fold expansion of functional HSCs [20], the expansion results in the production of both functional HSCs and non-HSCs. Cultured HSCs cannot be identified by the same panel of surface markers as freshly isolated HSCs. Endothelial Protein C Receptor (EPCR, also known as

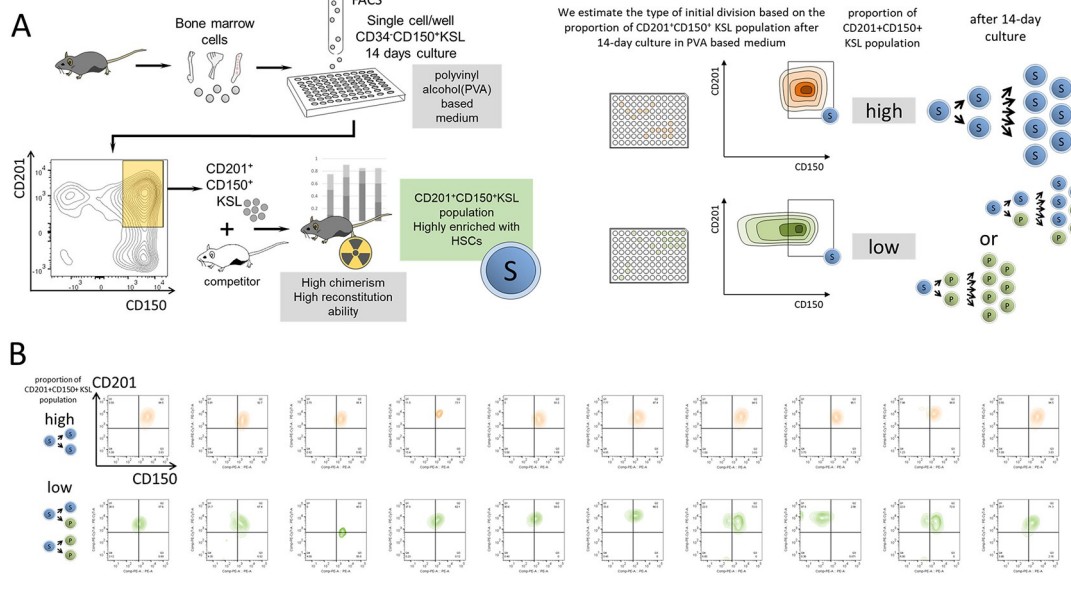

**Fig 1. Single cell expansion assays with PVA-based medium to examine the age-related changes of distribution of HSCs division modes. (A)** Overview of the *ex vivo* experiments and our previous report. Single-CD34⁻CD150⁺KSL cells sorted from mice of a certain week-old were cultured in wells with PVA-based medium. Using EPCR(CD201) and KSL as surface markers to identify cells after culturing in PVA medium, the cells show high repopulation ability in transplantation assay; this population is enriched with HSCs and we identified the division type based on the proportion of CD201⁺CD150⁺KSL cells after 14-day culture (Transplantation assays were performed in previous reports not in ours.) The proportion of CD201⁺CD150⁺KSL cells among whole expanded cells were measured and the mode of first cell division were inferred. **(B)** Some of the results of experiments on 7-week-old mice are shown as examples. Upper ten panel (shown in orange) represent wells with high CD201⁺CD150⁺KSL ratios after 14-day culture, and lower ten panel (shown in green) represent those with low CD201⁺CD150⁺KSL ratios. The first division is considered the S-S(orange), S-P, and P-P(green) divisions, respectively.

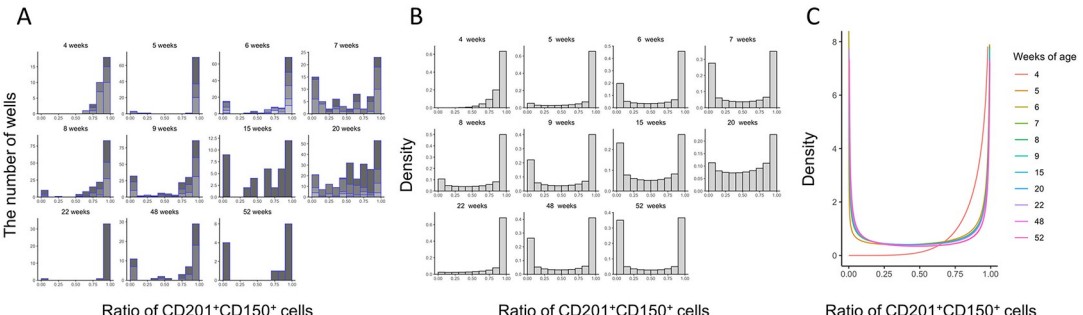

**Fig 2. Age-related changes of distribution of S-S, S-P, and P-P division quantified from single cell expansion assays. (A)** Histogram of the proportion of CD201⁺CD150⁺KSL population (CD201⁺CD150⁺KSL cells/total cells) after 14 days culture between 6 and 52 weeks. Different color bars are corresponding to the proportion from different mice. The number of mice used in the experiment at each week of age is as follows; 4 weeks, n = 3; 5 weeks, n = 3; 6 weeks, n = 5; 7 weeks, n = 3; 8 weeks, n = 3; 9 weeks, n = 3; 15 weeks, n = 1; 20 weeks, n = 4; 22 weeks, n = 1; 48 weeks, n = 3; 52 weeks, n = 1. **(B)** The beta distribution of the proportion of CD201⁺CD150⁺KSL population estimated by maximum likelihood estimation at each age by **Eq (1)**. The probabilities in 0.1 increments are shown in bar graphs. **(C)** The age-dependent distribution of the proportion of CD201⁺CD150⁺KSL population calculated from **Eq (2)**. The distribution curves of CD201⁺CD150⁺KSL population change from monomodal to bimodal distributions as age increase. It suggests that S-S division is predominant in young mice whereas P-P division gradually become apparent, showing bimodal peak. Our experiments suggested asymmetric division is not predominant between 6 and 52 weeks of age.

CD201) exclusively enriches for functional HSCs, as quantified by e*x vivo* and *in vivo* reconstitution assays [20, 26, 27]. The marker, with CD150, has also been recently shown to enrich for cells that possess long-term expansion potential and retain a transcriptional phenotype associated with HSCs after PVA-based culture. We, therefore, analyzed the proportion of functional HSCs within each well after *ex vivo* expansion based on CD201⁺CD150⁺ expression (**Fig 1B**). Next, we inferred the type of initial HSC division based on the proportion of functional HSCs (CD201⁺CD150⁺ KSL) after *ex vivo* expansion. We considered the three HSC divisions outlined in the literature (S-S division, S-P division, and P-P division). Wells containing a high proportion of CD201⁺CD150⁺ KSL cells arose from S-S divisions, and wells containing few CD201⁺CD150⁺ KSL cells arose from S-P or P-P divisions (**Fig 1A**).

We applied this single-cell expansion analysis to HSCs from mice of different ages. HSCs from young mice produced a unimodal peak of wells with high (e.g. 0.8 or above) CD201⁺CD150⁺ KSL cell proportions, indicative of cells undergoing S-S divisions (**Figs 1B and 2A**). HSCs from older mice produced a bimodal peak of wells with both low (e.g. 0.3 or below) and high (e.g. 0.8 or above) CD201⁺CD150⁺ KSL cell proportions, suggesting the occurrence of S-S and P-P divisions (**Figs 1B and 2A**). The cutoff lines, 0.3 and 0.8, are tentatively set numerical values, and exact estimation will be performed later through mathematical model analysis.

To quantitatively characterize the distributions of CD201⁺CD150⁺KSL cell proportions at each age, we assumed beta distributions and estimated the age-dependent distribution of CD201⁺CD150⁺KSL cell proportions (**Fig 2**, see **Methods, S1 Fig,** and **S1 Table**). In line with our qualitative observations, with age, the distribution of CD201⁺CD150⁺ KSL cell proportions becomes bimodal (**Fig 2**). These results provide a basis to characterize the age-related changes of cell division distributions.

## Age-related monotonical increase in murine HSC number

We hypothesized that the observed cell division changes in HSCs directly relate to HSC aging. Previous studies have suggested HSC numbers increase with age, but these reports have been restricted to limited time points or specific bones. To gain a better understating of the overall

process, we investigated how HSCs and progenitor cell numbers change with age across multiple time points and different bones.

To better capture the cell division dynamics and potential heterogeneity of HSCs *in vivo* (cycling vs non-cycling), we quantified total cell numbers and proportion of CD34$^-$CD150$^+$KSL cells (established marker for functional HSCs *in vivo*) from whole bone marrow and skeletal bones (femurs, pelvic bones, and sternums) freshly harvested from mice between 6 to 25 weeks of age (**Fig 3A**). Absolute total cell numbers did not significantly change within the whole bone marrow (**S2A Fig**). However, both the frequency and number of CD34$^-$CD150$^+$KSL cells increased with age (**Fig 3B** and **S2B Fig**). In contrast to HSCs, KSL and lineage-negative cells display unique, non-linear kinetics (**Fig 3C**; **S2C and S2D Fig**). To further validate our observations, we additionally sampled 40, 52, and 72-week-old mice. HSC numbers continued to increase linearly, while KSL and lineage-negative cells continued to demonstrate contrasting kinetics (**Fig 3B and 3C**; **S2C and S2D Fig**). We also analyzed the relationship between body weight and HSC or KSL cells but there is no clear correlation between those two factors (**S5 Fig**). Taken together, we demonstrate that aging induces a linear increase in total HSC numbers across the resident tissues sampled.

To understand how these age-related changes arise, we investigated the role of cell cycle and cell death. Dormant HSCs, especially in aged mice, might re-enter the cell cycle to repopulate the hematopoietic system and cause the HSC pool to expand. Alternately, with aging, HSCs may become less sensitive to apoptosis, thereby leading to the expansion of the HSC pool. First, using BrdU incorporation assay as a surrogate for cycling cells [28, 29], we compared *ex vivo* expanded HSCs from mice 6 to 20 weeks of age (**Fig 3D**). In line with previous reports, cell cycle rates of HSCs remained unchanged with age, suggesting a steady state of cell cycle [28, 29]. We next identified the proportion of annexin V-positive cells in the HSPC population from mice of different ages to compare changes in the fraction of apoptotic cells. Across all ages, the fraction of apoptotic HSCs remained negligible (**Fig 3E**). In summary, our data show that neither differences in cell cycle nor resistance to apoptosis are likely to be the cause of the increase in HSC numbers seen with aging. It is tempting to hypothesize that the changes in HSC division seen in our data likely explain this expansion.

## Quantifying age-related HSC division distribution

To test this hypothesis, we developed a mathematical model that utilized our *ex vivo* cell division data to quantify cell population dynamics. We defined two thresholds, $c_1$ and $c_2$, that mark the boundaries specifying the cell division type from the proportion of CD201$^+$CD150$^+$ KSL cells. These parameters are the thresholds for (a) S-S division where $x > c_2$, (b) S-P division where $c_1 < x < c_2$, and (c) P-P division where $x < c_1$. The fractions of each cell division type, S-S, S-P, and P-P division (denoted by $p(t)$, $q(t)$, and $1 - p(t) - q(t)$, respectively, see **Methods**) can be then calculated as the fraction of cells within the learnt boundaries of each respective division from the *ex vivo* data in **Fig 1**. We combined these mathematical implementations of the fractions of cell divisions with a population dynamics model considering cell fluxes to obtain a mathematical model that describes the dynamics of HSC differentiation with the age-dependent HSC divisions (**Fig 4A and Eqs (5 and 6)** in **Methods**). We fitted this model to the time-course numbers of HSCs and KSL cells (obtained from the *in vivo* experiments in **Fig 3**) to estimate parameters in **Eqs (5 and 6)** for the dynamics of cell differentiation (initial population sizes and cell cycle rate of HSCs) and the cell division type thresholds ($c_1$ and $c_2$) (see **Methods**).

We found our mathematical model was able to capture the monotonic increase of HSC and the irregular transitions of the KSL population (**Fig 4B**). The estimated threshold values, $c_1$

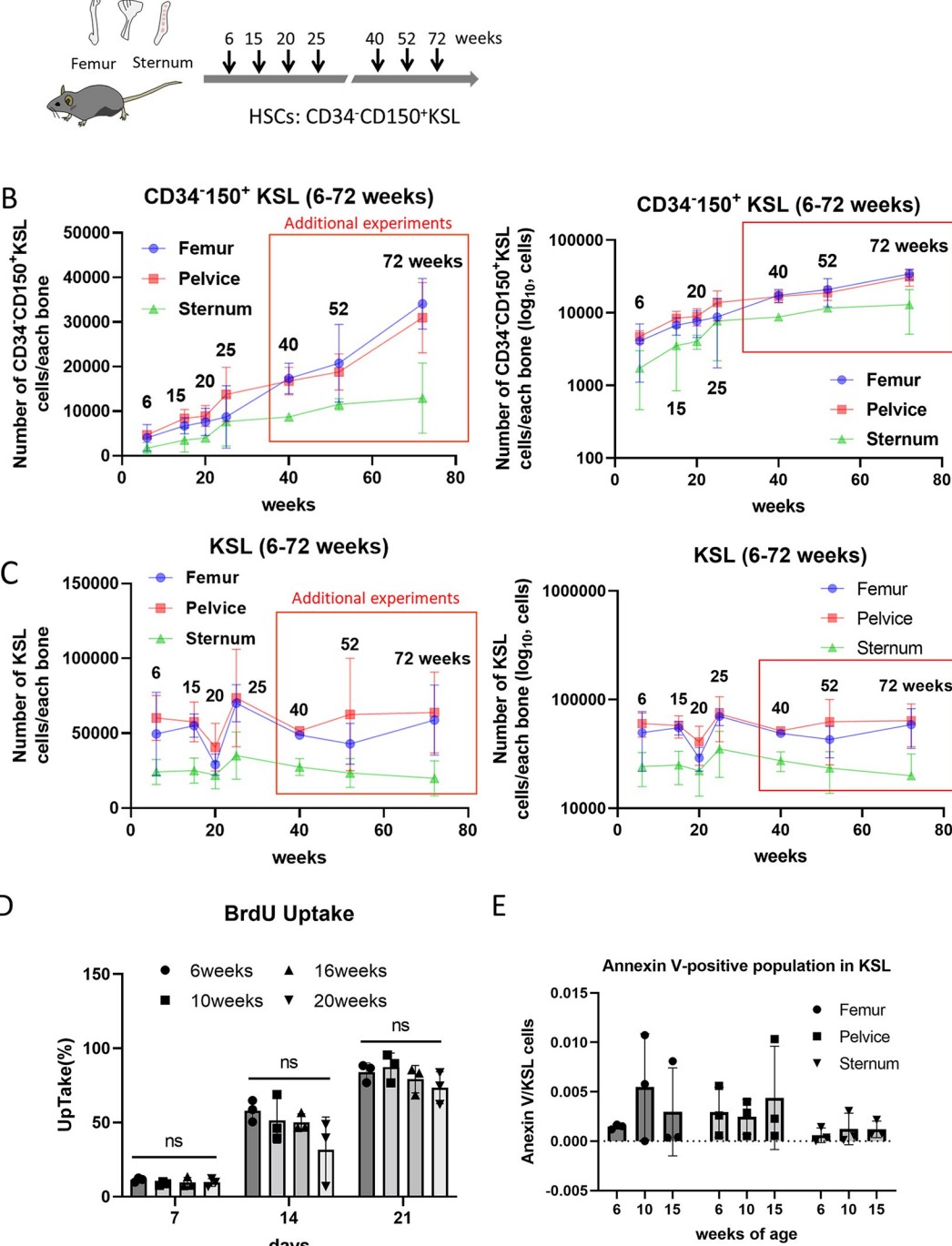

**Fig 3. Age-related qualitative changes of HSC and KSL cell dynamics. (A)** Overview of the *in vivo* experiments. The numbers of CD34-CD150+HSCs and KSL cells in bone marrows of femur, pelvice and sternum were measured from 6, 15, 20, 25, 40, 52, and 72 weeks-old mice. Three mice were used for the experiments of each week of age. **(B)** The number of CD34-CD150+HSCs population of bone marrow in each bone between 6 and 72 weeks of age. Dots and error bars are averages and standard deviations. The data in the red box were separately measured for our validation. The right panel represents the same data as the left panel, but the vertical axis was transformed to $\log_{10}$. **(C)** The number of KSL population of bone marrow in each bone between 6 and 72 weeks of age. Dots and error bars are averages and standard deviations. The data in the red box were separately measured for our validation. The right panel represents the same data as the left panel, but the vertical axis was transformed to $\log_{10}$. **(D)** Bromodeoxyuridine (BrdU) uptake assay between 6 and 20 weeks. There were no significant differences depending on age. The data were analyzed by one-way analysis of variance (ANOVA). If not

otherwise indicated, each data point represents one independent mouse. **(E)** Annexin V-positive population in KSL at age 6, 10 and 15 weeks. Three mice were used for the experiments of each week of age. Annexin V-positive population in KSL was 0.01 or less in all ages, which was negligibly small. If not otherwise indicated, each data point represents one independent mouse.

and $c_2$, were 0.3 and 0.8, respectively, and HSC cell cycle rate was inferred as 0.172 per week, in line with previous estimates (**Table 1**). The expected HSC division distribution is shown in **Fig 4C**. Our model inferred that symmetric self-renewal, S-S division, is dominant at a young age (52.4% at 10 weeks) and its fraction decreases with age (41.6% at 70 weeks), whereas the fraction of symmetric differentiation, P-P division, increases with age (24.9% and 41.5% at 10 and 70 weeks, respectively) (**Fig 4C; S3** and **S4 Figs**). The fraction of asymmetric division, S-P division, is relatively low throughout all ages (16.9–22.7%). Taken together, by assigning data obtained from *ex vivo* culture experiments to HSC division patterns and extracting information from *in vivo* cell number changes through a novel mathematical model, we have succeeded in quantitatively understanding how HSC division changes with age.

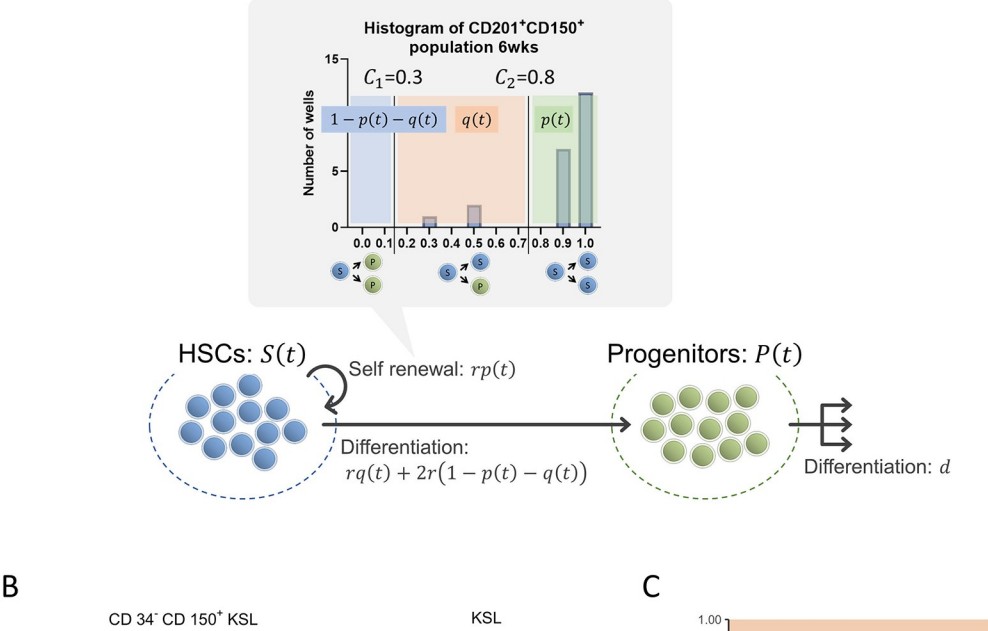

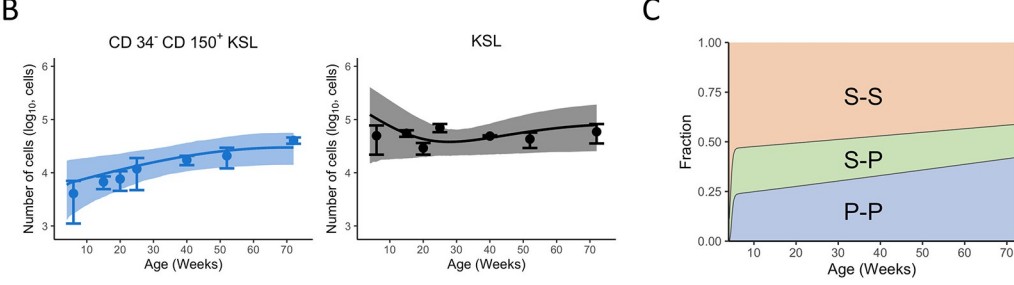

**Fig 4. Age-dependent distribution of three division modes of HSCs. (A)** Overview of the mathematical model describing the dynamics of HSC differentiation. HSCs proliferate by self-renewal and differentiate into the progenitor population. The types of HSC division are defined by the proportion of CD201+CD150+KSL cells. Progenitor cells decrease due to the differentiation into more differentiated cells and mortality. **(B)** The age-related number of CD34-CD150+KSL cells (HSCs, left) and KSL cells (HSCs and progenitor cells, right) in femur predicted by the mathematical model (**Eqs (5)** and (**6**)) with best-fit parameters and the experimental data are shown. The lines and filled area are corresponding to the mean and 95% credible interval, respectively, and the dots and error bars are the average and standard deviations. **(C)** The expected age-dependent distributions of S-S, S-P, and P-P divisions in mice were calculated.

## Discussion

Organismal aging, especially its correlation with stem cell aging has intrigued the scientific community for decades. In stem cells, both extracellular and intracellular factors regulate cellular processes (like cell division), eventually regulating the aging process [18, 30–32]. The aging of HSCs, primarily owing to its association with clonal hematopoiesis is of utmost importance, as it is associated with a plethora of pathogenic disease states [33, 34]. Here, we combine quantitative mathematical modeling with wet-lab experiments to demonstrate the dynamics of age-related changes in the poorly understood HSC (a) S-S division, (b) S-P division, and (c) P-P division.

We build upon and expand our understanding of previous reports showing an increase in HSC pools with age [6, 7, 13, 18, 35]. We demonstrate the size of the HSC population linearly increases with age in mice. Furthermore, we show that the observed kinetics are consistent across multiple HSCs resident tissues (like femur, pelvis, and sternum). Interestingly, our mathematical model and experimental data show an unexpected temporary drop in the KSL population at ~20 weeks of age. This is consistent with the onset of age-related bone loss. Some previous studies reported that peak bone mass is achieved at 4–6 months of age and declines thereafter in mice [35, 36]. In human, the decline in bone mass begins at start of the 4th decade, while age-associated alterations come into play later, at or after the 5th decade [37]. In other words, the onset of bone loss, preceding other hallmarks of aging, would have some relationship with KSL unique kinetics. Further in-depth analysis is required to understand the biological significance of this observation.

In recent years, studies have begun to explore the age-related changes of HSC division type. Arai et al. combined single-cell profiling with machine learning to predict the division type of HSCs taken from mice of three different ages [14]. They report that in young mice (4 weeks old), S-S division is predominant (accounting for 50–60% of all divisions), followed by S-P and then P-P division. In adult mice (8 weeks old), they find all three division types to be equally distributed, whereas in aged mice (18 months old), they find the divisions to be almost exclusively P-P division [14]. Our work, across 11 ages, supports their observation of S-S division dominance in young mice, and increasing cell tendency for P-P division in older mice. This finding would be valid; many previous reports demonstrate that young mice exhibit high self-renewal and regenerative capacities, but HSCs are vulnerable to aging-related stress and gradually lose these capacities [38–40]. As mentioned above, most of the experimental results previously reported agree that the age-related increase of HSCs pool size [6, 7, 13, 18, 35], and so it is reasonable to think that S-S divisions play a dominant role to expand the HSCs pool in young age. Some previous studies suggested one model for aging-related changes in which HSCs in young mice undergo asymmetric division but switch to symmetric division during aging [41–43]. This idea is inconsistent with active proliferation of HSCs or self-renewal capacity of HSCs in young mice.

We need more discussion on cell division kinetics of older mice. Our experiments suggest that age-related alteration occurred in HSCs division modes and were consistent with previous reports. If S-S divisions predominate until late in murine life, increasing of HSCs pool size would not show linear but exponential growth kinetics. Nor is it consistent with age-dependent loss of self-renewal capacity. There are several theories to explain age-related alteration of HSCs division mode. As we mentioned above, Arai et al. observed almost exclusive P-P divisions in older mice (18-month-old mice) [14]. M. Carolina Florian et al. report around 65% of HSCs divide symmetrically in 20-26-month-old mice (they did not mention the ratio of S-S and P-P divisions) [44]. Some reports argue that HSCs in cell cycle undergo asymmetric divisions in most of the time [7], while others demonstrate that symmetric self-renewal regulates

the age-related HSCs pool size expansion [44, 45]. The model mentioned above demonstrate HSCs undergo asymmetric division and switch to symmetric division in aged HSCs. Among them, some hypotheses that argue that P-P and/or S-P divisions are predominant in later life were inconsistent with known facts, age-associated expansion of HSCs pool size. In order to maintain the stem cell pool, the self-renewal ratio of the HSC has to be more than 50% [46]. If P-P and/or S-P divisions are preferentially undergone over S-S divisions, HSC pool size would be contracted. The difference between observations may be attributed to our novel approach using PVA-based medium which can reproduce the in vivo environment *ex vivo* better than before. As we mention later, differences in assumptions and models may also contribute to differences in results.

However, the dynamics of HSCs remain unclear at very old ages and if the HSC pool size decreases in older mice, P-P divisions could become increasingly dominant. In our model, the self-renewal ratio of the HSCs accounts for 50% or less in old age (60–70 weeks) and the increase of the number of HSCs reached a plateau.

Recent studies have also modelled the age-related increase in the HSC pool within a short time period [19]. These models infer HSC flux rates, death rates, division rates and division type distributions from fate-mapping and cell cycle data. Based on these model parameters, Barile et al. report most divisions to be predominantly S-S divisions; at least four out of five HSC divisions are inferred to be symmetric self-renewal [19]. These differences could arise from modelling assumptions such as (a) all HSCs differentiate into multipotent progenitors, or (b) there is no P-P division. We and others [47] show that HSC division and distributions change with age. As such, merging our methods of age-dependent mathematical modeling with the methods of estimation by fate-mapping could allow better insights into HSCs and reconcile these existing disparities.

We mathematically interpreted the distribution of the proportion of CD201+CD150+ KSL obtained ex vivo by approximating it with a beta distribution and estimated the age-related changes in the fraction of division patterns by estimating the thresholds that separate division patterns of HSCs from the changes in the number of cells obtained in vivo. In this mathematical formulation, we made assumptions that the distribution of the proportion of CD201+CD150+ KSL follows a beta distribution and that its shape parameters follow an biexponential function. This is one limitation of this study and should be investigated including biological interpretations. However, this formula was sufficient for the sole purpose of mathematically expressing the age-related changes in the distribution of the proportion of CD201+CD150+ KSL. In fact, by incorporating the fractions of division patterns derived from this distribution of the proportion of CD201+CD150+ KSL, our mathematical model for the differentiation of HSCs into progenitor cells could represent changes in cell number in mice.

In conclusion, our work answers two long-standing questions in the field (a) Symmetric self-renewal or S-S division of HSCs is the predominant mode of division throughout the lifespan of an animal, and its fraction gradually decreases with age. (b) HSC pool size increase with age across relevant tissue compartments. These findings have the potential to help elucidate the mechanisms governing clonal hematopoiesis. HSCs in adults are mostly in a quiescent state and only get activated (to expand and differentiate) in response to stimuli from infection, inflammation, chemotherapy, radiation, bone marrow transplantation, and aging, among others [48–52]. S-S division has been suggested as the underlying cause regulating the expansion of HSC pools in these settings [49, 50, 52]. One potential limitation of our study is that we have not looked into how HSC divisions are regulated. Recent reports suggest that the HSC niche plays a major role and aging is a critical regulator of this process [14, 53, 54]. Further investigation into deciphering the mechanism shall allow controlling HSC division types, and eventually HSC pool size. Such insights will prove fundamental in controlling the cell

dynamics driving age-related diseases such as clonal hematopoiesis. Thus, it is tempting to speculate how applying our novel hybrid approach to mice models (like transgenic Tet2 or Dnmt3a) will help elucidate the novel mechanism underlying clonal hematopoiesis, aging among others. In summary, our study paves the way for an understanding of the aging process and lays the foundations to elucidate the mechanisms underlying HSC biology.

## Supporting information

**S1 Table. The parameter values of beta distribution estimated by the likelihood estimation from the proportion of CD201$^+$CD150$^+$KSL population.** The estimated values of shape parameters, $\alpha_t$ and $\beta_t$, for each age.
(PDF)

**S2 Table. The estimated parameter values of Eqs 3 and 4.** Based on the estimated shape parameters, $\alpha\_t$ and $\beta\_t$, for each age (shown in S1 Table), the above parameters in Eqs (3 and 4) were estimated by a nonlinear least squares method (S1 Fig). The estimated parameter values of Eqs (3 and 4) are shown.
(PDF)

**S3 Table. List of antibodies, table of anti-mouse antibodies used in this study including supplier and identifier. (A)** Antibodies used to collect HSCs FACS. **(B)** Antibodies used for the analysis of 14-day culture of HSC. **(C)** Antibodies used for the analysis of age-related murine HSC number.
(PDF)

**S1 Fig. Age-dependent parameters of beta distribution. (A)** The age-dependent shape parameters of beta distributions of CD201$^+$CD150$^+$KSL population, $\alpha(t)$ and $\beta(t)$, were estimated. The dots and lines are corresponding to the values of shape parameters, $\alpha_t$ and $\beta_t$, estimated by maximum likelihood estimation from data with each age and **Eqs (3)** and **(4)**, respectively. **(B)** Each figure shows the combination of the histogram of the proportion of CD201+CD150+KSL population (shown in Fig 2A) and the line derived by maximum likelihood estimation.
(TIF)

**S2 Fig. Age-related alterations of HSCs and progenitor cells. (A)** The total numbers of nucleated bone marrow cells between 6 and 72 weeks of age in each bone was shown. There was no significant difference as age increased (femur, $p = 0.824$; pelvic, $p = 0.324$; and sternum, $p = 0.870$). The data were analyzed by one-way analysis of variance (ANOVA). If not otherwise indicated, each data point represents one independent mouse. **(B)** The frequencies (left) and numbers (right) of CD34$^-$CD150$^+$HSC between 6 and 72 weeks of age in bone marrow in each bone were shown. **(C)** The frequencies (left) and numbers (right) of KSL cells between 6 and 72 weeks of age in bone marrow in each bone were shown. **(D)** The frequencies (left) and numbers (right) of lineage-cells between 6 and 25 weeks of age in bone marrow in each bone were shown. They are similar to those of KSL cells.
(TIF)

**S3 Fig. Identification of $c_1$ and $c_2$, and sensitivity analysis of the differentiation rate of progenitor cells. (A)** The probabilities in 0.1 increments for age-dependent distributions of CD201$^+$CD150$^+$HSC population calculated from **Eq (10)** and the estimated boundaries of cell division types (i.e., the two thresholds: dashed lines) are shown. **(B-E)** The results of model fitting to the cell count data in the same procedure of **Fig 4B and 4C** with differentiation rates of progenitor cells, $d$, fixed as 0.05 **(B, D)** and 0.2 **(C, E)**, respectively. Dots and error bars in **(B)**

and **(C)** are the averages and standard deviations of the experimental data. The lines and filled area in **(B)** and **(C)** are the mean and 95% credible intervals predicted by the mathematical model. **(D, E)** The corresponding age-dependent distributions of S-S, S-P, and P-P divisions in mice were calculated.
(TIF)

**S4 Fig. Age-dependent change of the distribution of cell divisions.** The expected distributions of S-S (red), S-P (green), and P-P (blue) divisions in mice at the specific ages 10, 20, 30, 40, 50, 60 and 70 weeks were showed from the calculation in **Fig 4C**.
(TIF)

**S5 Fig. The relationship between body wight and HSC number or frequencies.** The upper panel shows the relationship between the age and body weight of mice. Three male mice were used in each age. The lower panel shows the correlation coefficient between body weight and frequencies or the number of CD34-CD150+KSL population. There is no significant correlation.
(TIF)

**S1 Data. Source data Figs 1B and 2.**
(XLSX)

**S2 Data. Source data Fig 3E.**
(XLSX)

**S3 Data. Source data of extended data Fig 4.**
(PDF)

**S4 Data. Source data of Fig 2B and extended Fig 3.**
(XLSX)

## Acknowledgments

We thank the University of Tokyo Institute of Medical Science (IMSUT) FACS core laboratory for expert technical assistance.

## Author Contributions

**Conceptualization:** Munetomo Takahashi, Shingo Iwami.

**Data curation:** Teiko Kawahigashi, Shingo Iwami.

**Formal analysis:** Teiko Kawahigashi, Shoya Iwanami, Shingo Iwami.

**Investigation:** Teiko Kawahigashi, Shingo Iwami.

**Project administration:** Teiko Kawahigashi.

**Resources:** Shoya Iwanami.

**Supervision:** Shingo Iwami, Satoshi Yamazaki.

**Writing – original draft:** Teiko Kawahigashi, Munetomo Takahashi, Joydeep Bhadury, Satoshi Yamazaki.

**Writing – review & editing:** Teiko Kawahigashi, Joydeep Bhadury, Satoshi Yamazaki.

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
