## [Decision Letter · Decision Letter 0]

21 Nov 2023

PONE-D-23-30991Age-related changes in the hematopoietic stem cell pool revealed the balance of symmetric and asymmetric divisionsPLOS ONE

Dear Dr. Yamazaki,

Thank you for submitting your manuscript to PLOS ONE. After careful consideration, we feel that it has merit but does not fully meet PLOS ONE’s publication criteria as it currently stands. Therefore, we invite you to submit a revised version of the manuscript that addresses the points raised during the review process.

Please submit your revised manuscript by Jan 05 2024 11:59PM. If you will need more time than this to complete your revisions, please reply to this message or contact the journal office at plosone@plos.org. Please include the following items when submitting your revised manuscript:A rebuttal letter that responds to each point raised by the academic editor and reviewer(s). You should upload this letter as a separate file labeled 'Response to Reviewers'.A marked-up copy of your manuscript that highlights changes made to the original version. You should upload this as a separate file labeled 'Revised Manuscript with Track Changes'.An unmarked version of your revised paper without tracked changes. You should upload this as a separate file labeled 'Manuscript'.If applicable, we recommend that you deposit your laboratory protocols in protocols.io to enhance the reproducibility of your results. Protocols.io assigns your protocol its own identifier (DOI) so that it can be cited independently in the future. For instructions see: https://journals.plos.org/plosone/s/submission-guidelines#loc-laboratory-protocols. Additionally, PLOS ONE offers an option for publishing peer-reviewed Lab Protocol articles, which describe protocols hosted on protocols.io. Read more information on sharing protocols at https://plos.org/protocols?utm_medium=editorial-email&utm_source=authorletters&utm_campaign=protocols.

We look forward to receiving your revised manuscript.

Kind regards,

Zoran Ivanovic, MD, PhD, HDR

Academic Editor

PLOS ONE

Journal Requirements:

"none"

5. Please amend the manuscript submission data (via Edit Submission) to include authors Dr. Shoya Iwanami, Dr. Munetomo Takahashi, Dr. Joydeep Bhadury, and Dr. Shingo Iwami.

7. We notice that your supplementary [S1-S4 Figure] are included in the manuscript file. Please remove them and upload them with the file type 'Supporting Information'. Please ensure that each Supporting Information file has a legend listed in the manuscript after the references list.

Additional Editor Comments:

As a phenotypic HSC "surrogate" , the population KSL CD201+CD150+ population has been used. Please discuss the possiblity of the "dissociation phenotype-function ex vivo". Also you may provide some arguments that the KSLCD201+CD150+ population can be considered as a bone fide approximation of the functional HSCs.

Reviewers' comments:

Reviewer's Responses to Questions

**Comments to the Author**

1. Is the manuscript technically sound, and do the data support the conclusions?

Reviewer #1: Yes

2. Has the statistical analysis been performed appropriately and rigorously? 

Reviewer #1: I Don't Know

3. Have the authors made all data underlying the findings in their manuscript fully available?

Reviewer #1: Yes

4. Is the manuscript presented in an intelligible fashion and written in standard English?

Reviewer #1: Yes

5. Review Comments to the Author

Reviewer #1: In the present paper, the authors developed mathematical models to analyze and describe HSC division types across aging in mouse. For this purpose, they used ex vivo data to define parameters linked to HSC division type and in vivo data to estimate parameters for cell differentiation dynamics. The authors showed through single cell assay and analysis of CD201+ CD150+ KSL proportion after ex vivo culture, an age-related change in HSC division type. Indeed they observed a high rate of symmetric self-renewal division in HSC coming from young mice, and progressive increase of asymmetric self-renewal and symmetric differentiation divisions with increase of mouse age. In vivo analysis of CD34- CD150+ KSL total number and frequency highlighted an age-related increase of this population, which was not associated to cell cycle rate modification nor apoptosis resistance. Mathematical modeling was subsequently performed and properly described age-related HSC and progenitors dynamics observed in vivo. Finally, the authors concluded that symmetric self-renewal divisions predominantly occurs in HSCs during animal’s lifespan.

Age-related HSCs division mode is an intensive research area. The authors present very interesting data that are in contrast with several studies already published, where young HSCs were associated with asymmetric self-renewal and symmetric self-renewal was related to aged HSCs. The data presented in this article are convincing and well-argued and give food for thought on this crucial subject.

I am favorable to the publication of this article with the minor revisions below :

1 – Results, lines 146/147 : the statement is associated with figure 4A but probably should be associated with figure 4B.

2 – Results, line 148 : description of the different parameters could be relied to Table 1.

3 – Results, line 149 : the expected HSC division distribution in presented in figure 4C, but is associated with figure 4B in the text.

4 – Discussion, line 198 : reference n°41 is not related to the statement line 196-197.

5 – Materials and methods, lines 270-276 : several chemicals are proposed to be tested in culture while text and figure1A show the selective use for PVA during the experiments.

6 - Materials and methods, lines 285-289 : antibodies used for HSC caracterization and their manufacturer’s origin could be detailed.

7 - Materials and methods, line 294 : the name of the post-hoc test is not specified.

8 – Figure 1B : upper pannel n°7 (from the right side) correspond to middle proportion of CD201+ CD150+ KSL, but is classified in high proportion of CD201+ CD150+ KSL

9 - Figure 1B : contrary to the figure 1B legend, no pannel for low CD201+ CD150+ KSL ratio (purple) is proposed.

10 – Figure 2A : I don’t understand why the sum of all frequencies for a graph is not closed to 100. Is the ordinate value represents number of wells instead of frequency ?

11 – Figure 2A : in the graph corresponding to mice aged of 6 weeks, the last bar (ratio 0,9-1) shows 5 parts while it is given for 4 mice in the legend. Is there 5 mice for this age?

12- Figure 3B and C : maybe the authors could precise in the legend that the right pannel represent the same data compared to left pannel, with log10 transformation of data.

13 – Figure S1B : there is no description in the legend for this part.

14 – Figure S5 : there is no legend provided for this figure, and it is not quoted in the text.

6. PLOS authors have the option to publish the peer review history of their article (what does this mean?). If published, this will include your full peer review and any attached files.

Reviewer #1: No

---

## [Editor Report · Decision Letter 1]

3 Jan 2024

Age-related changes in the hematopoietic stem cell pool revealed via quantifying the balance of symmetric and asymmetric divisions

PONE-D-23-30991R1

Dear Dr. Yamazaki,

We’re pleased to inform you that your manuscript has been judged scientifically suitable for publication and will be formally accepted for publication once it meets all outstanding technical requirements.

Kind regards,

Zoran Ivanovic, MD, PhD, HDR

Academic Editor

PLOS ONE
---

## [Editor Report · Acceptance letter]

18 Jan 2024

PONE-D-23-30991R1 

PLOS ONE

Dear Dr. Yamazaki, 

I'm pleased to inform you that your manuscript has been deemed suitable for publication in PLOS ONE. Congratulations! Your manuscript is now being handed over to our production team.

Kind regards, 

on behalf of

Dr. Zoran Ivanovic 

Academic Editor

PLOS ONE